# Role of Circulating Biomarkers in Platinum-Resistant Ovarian Cancer

**DOI:** 10.3390/ijms222413650

**Published:** 2021-12-20

**Authors:** Carolina Maria Sassu, Innocenza Palaia, Serena Maria Boccia, Giuseppe Caruso, Giorgia Perniola, Federica Tomao, Violante Di Donato, Angela Musella, Ludovico Muzii

**Affiliations:** Department of Maternal and Child Health and Urological Sciences, “Sapienza” University of Rome, Polyclinic Umberto I, 00161 Rome, Italy; carolinamsassu@hotmail.it (C.M.S.); bocciaserena@gmail.com (S.M.B.); g.caruso@uniroma1.it (G.C.); giorgia.perniola@uniroma1.it (G.P.); federica.tomao@uniroma1.it (F.T.); violante.didonato@uniroma1.it (V.D.D.); angela.musella@uniroma1.it (A.M.); ludovico.muzii@uniroma1.it (L.M.)

**Keywords:** platinum-resistant ovarian cancer, circulating biomarker, liquid biopsy, prognosis, drug response biomarker

## Abstract

Ovarian cancer (OC) is the second most common cause of death in women with gynecological cancer. Considering the poor prognosis, particularly in the case of platinum-resistant (PtR) disease, a huge effort was made to define new biomarkers able to help physicians in approaching and treating these challenging patients. Currently, most data can be obtained from tumor biopsy samples, but this is not always available and implies a surgical procedure. On the other hand, circulating biomarkers are detected with non-invasive methods, although this might require expensive techniques. Given the fervent hope in their value, here we focused on the most studied circulating biomarkers that could play a role in PtR OC.

## 1. Introduction

Ovarian cancer (OC) is the second most common cause of death in women with gynecological cancer, with 313,959 new cases and 207,252 deaths in 2020 around the world [1]. High-grade serous ovarian cancer (HGSOC) is the most common form of OC (about 70% of epithelial ovarian cancer EOC) [2] and is characterized by high mortality due to diagnosis at an advanced-stage disease in about 75% of cases [3]. After a positive response to upfront treatment, OC recurs in the majority of patients and develops progressive resistance to therapy, limiting effective treatment options. According to the Gynecologic Oncology Group (GOG), recurrent ovarian cancer has been classified on the basis of platinum-free interval (PFI) between last platinum administration and recurrence [4]. However, this classification is just arbitrarily defined. Firstly, the time to recurrence depends on the timing and methods of the follow-up assessment. Secondly, maintenance therapy retards relapse, and this inevitably revolutionizes the concept of platinum sensitivity [5].

Consequently, the last Gynecologic Cancer Intergroup consensus conference proposed to refer to the therapy-free interval (TFI) [6], but this change is still hard to be applicable in clinical practice. Thus, referring to the historic classification based on PFI, a relapsed disease that occurs within 6 months of the last administration of platinum is classified as platinum-resistant (PtR) and represents the greatest challenge for specialists and researchers [4]. PtR disease is, in fact, correlated to poor prognoses with a low response rate (<20%) to subsequent lines of therapy and reduced progression-free survival (PFS) (about 4 months), as well as median overall survival (OS) (<12 months) [7,8].

Considering the poor prognosis of OC, especially HGSOC, huge effort was made to pinpoint new predictive biomarkers able to help physicians in approaching and treating OC patients. The World Health Organization defines a biomarker as “any substance, structure or process that can be measured in the body or its products and influence or predict the incidence of outcome or disease” [9]. In general, cancer biomarkers include every compound present in or produced by cancer cells or by other cells of the organism in response to and in correlation with the tumor [10]. Biomarkers can be measured accurately and reproducibly [11] in the blood, urine, stool, or other fluid (circulating markers) or in tumor samples (tissue biomarkers). The former group is represented by surface antigens, proteins/lipids, nucleic acids (DeoxyriboNucleic Acid [DNA] and RiboNucleic Acid [RNA]), hormones, circulating cancer cells, and inflammatory ones. Specifically, cancer biomarkers may have several potential usages (diagnostic, prognostic, and predictive value) [12,13]. However, very often, the same biomarker presents various overlapping roles, thus defining a unique clear role could be challenging and not always possible, as shown by the intersection of sets in Figure 1.

Currently, most biomarkers derive from tumor biopsy samples. Nevertheless, it is not always possible to perform surgery in order to obtain a biopsy due to the risk of complications in frail patients, such as bleeding and infections, and the difficulty of surgical procedures in some organs. Moreover, given the mutagenicity of the disease, referring to a previous histological specimen may not reveal details of the actual tumor status. Finally, a single tissue sample does not always provide exhaustive data of the tumor genome (due to sampling bias). Recently, therefore, circulating biomarkers have raised more interest thanks to the advantage of being detected with a non-invasive method, along with a better benefit-cost ratio. Characteristics of circulating biomarkers and tissue biopsy are shown in Table 1. Unfortunately, most of these serum biomarkers are not sufficiently sensitive and specific to make screening and early diagnosis in the general population.

This review analyses and reports data from studies on circulating biomarkers with a potential prognostic and predictive function in patients with PtR OC.

## 2. Methods

A search in PubMed up to June 2020 was performed, combining the following terms: “circulating biomarkers”, “platinum resistant ovarian cancer”, “liquid biopsy”, “genetic and epigenetic”, “inflammation”, and “angiogenesis” reveal published evidence in the last 25 years. Unpublished or non-peer-reviewed studies, papers without available full-text and non-English manuscripts were excluded.

## 3. Circulating Biomarkers

Table 2 and Figure 2 resume circulating biomarkers discussed in this review. Details will be extrapolated in the following sections.

### 3.1. Glycoprotein Biomarkers

CA125 and HE4 are the only validated circulating biomarkers approved for the OC diagnosis [14].

#### 3.1.1. CA125

CA125/mucin 16 (MUC16) is a member of the mucin family glycoproteins encoded by the MUC16 gene. It promotes cancer cell proliferation and inhibits anti-cancer immune responses. Serum CA125 is also a prognostic marker used to predict OC patient survival [15]. Moreover, it was shown to be a predictor of response to chemotherapy [16]. Results from a trial that investigated the role of CA125 in regulating the sensitivity of epithelial OC cells to different types of genotoxic drugs revealed that CA125 promotes cisplatin resistance. In particular, this effect seems to be mediated by the C-terminal domain (CTD) of CA125. Experimental overexpression of this domain in CA125 negative OC cells confers platinum resistance, while the downregulation of CA125, mediated by CA125-specific single-chain antibodies that prevent its localization in the cell surface, increases by approximately 5 times cisplatin cytotoxicity, promoting cisplatin-induced apoptosis [17].

Additionally, serum CA125 dosage combined with the ascites concentration of an inflammatory biomarker such as leptin seems to be able to predict prognosis and response to treatment in OC patients. Serum CA125/ascites leptin ratio was found to be a predictor of resistance to first-line platinum-based therapy (*p* = 0.02) and poor outcomes in terms of PFS (*p* = 0.04) and OS (*p* = 0.04) in patients with OC [18]. Finally, CA125 levels were incorporated in a nomogram to predict the probability of 1-year OS and median survival in patients with PtR OC. The CA125 has proved to have a relevant prognostic significance (contributing 13 points out of 100) after performance status (38 points), ascites (19 points), and size of largest tumor documented on imaging (14 points) [19]. However, the real application of CA125 dosage is still limited in clinical practice. In fact, the increase of CA125 concentration without symptoms of a disease does not legitimate an immediate initiation of chemotherapy after complete response to first-line platinum since the evidence showed that early treatment has no survival benefit [20].

#### 3.1.2. HE4

Human Epididymis Protein 4 (HE4) is a secretory glycoprotein, a member of the family of acidic four-disulfide core proteins. It is expressed in both the male and female reproductive tract and other normal human tissues such as the breast, kidney, respiratory tract and is highly overexpressed in epithelial ovarian cancer. Recently, HE4 has also been detected in OC patients’ urine [21].

Several studies demonstrated that serum HE4 levels were higher in platinum-resistant OC patients and that HE4 promotes platinum resistance both in vitro and in vivo [22,23]. However, the way in which it promotes platinum resistance is not clear. The most likely hypothesis is that multiple mechanisms may play a role in HE4-mediated chemo-resistance. Early Growth Response gene 1 (EGR1), a mitogen-activated protein kinase (MAPK)-regulated transcription factor involved in promoting apoptosis, was induced by several factors such as platinum compounds. HE4 overexpression seems to suppress cisplatin-mediated upregulation of EGR1 [22]. Angioli et al. showed that serum HE4 levels during first-line chemotherapy predict platinum-resistant disease at the third chemotherapy cycle with 100% sensitivity and 85% specificity. Furthermore, they also reported that CA125 levels during chemotherapy were not statistically significant in predicting platinum response [24].

#### 3.1.3. Mesothelin

Mesothelin (MSLN), a glycosylphosphatidylinositol (GPI) anchored cell surface protein, is physiologically expressed on mesothelial cells and is overexpressed in several types of tumors, including ovarian cancer. The MSLN gene maps on chromosome 16p13.3 and encodes a protein precursor of 71 KDa, proteolytically cut in a C-terminal fragment of 40 KDa (so-called mesothelin), bound to the cell membrane by a glycosyl-phosphatidylinositol and in an N- terminal fragment of 31 KDa (so-called MPF), secreted in the serum, with a megakaryocyte-enhancing action [25].

The soluble form of MSLN appears to arise through alternative splicing of the MSLN gene that disrupts the GPI-anchor motif. Another hypothesis suggests that soluble MSLN may be a cleavage product of the membrane-bound MSLN. Studies have shown that several mechanisms exist in which MSLN plays a role in cell adherence, cancer progression, and chemoresistance. It has been suggested that MSLN can bind to CA125 to mediate cell adhesion aiding in the peritoneal implantation and metastasis process [26]. Moreover, it may promote cancer cell survival and proliferation via the Nuclear Factor kappa-light-chain-enhancer of activated B cells (NF-Kb) signaling pathway and seems to confer resistance to cytotoxic drug-induced apoptosis, down-regulating the pro-apoptotic protein Bim [27].

Cheng et al. reported that PtR OC patients showed significantly higher MSLN expression on cancerous tissue specimens than chemo-sensitive patients (*p* < 0.001) and that its expression is associated with worse PFS (*p* = 0.03) and OS (*p* < 0.008) of OC patients [28].

However, elevated MSLN levels were also found in the serum and urine of OC patients with early (*p* = 0.02) and late (*p* < 0.001) disease. Urinary MSLN dosage also showed good sensitivity for early-stage OC (42%) compared to other markers tested in the study [29], although not proved insufficient for an effective screening strategy. Further studies suggest the potential use of this soluble glycoprotein as a circulating diagnostic marker for OC [30]. Anyway, its diagnostic value in OC, and especially in PtR disease, is not yet satisfactory.

### 3.2. Liquid Biopsy

Recently, the role of “liquid biopsy” in cancer has been attracting more attention. Strictly speaking, this term refers to the analysis of circulating cell-free DNA (cfDNA) and circulating tumor cells (CTCs), expanding the spectrum of other circulating biomarkers already used in clinical practice. The detection and analysis of these compounds in patients’ blood may represent a non-invasive tool to obtain information for the diagnosis, prognosis, and monitoring of tumor genotype in OC [31,32]. Moreover, through liquid biopsy, extracellular vesicles (EVs) (including exosomes, microvesicles, and other membranous structures) and circulating cell-free microRNAs (cfmiRNAs) can also be detected (Figure 3). miRNAs, which are responsible for epigenetic alterations, are discussed in a separate section.

#### 3.2.1. Circulating Tumor DNA

The detection of circulating cfDNA in biological fluids is a physiological phenomenon caused by DNA released from apoptotic or necrotic cells. In cancer patients, this circulating cfDNA also originates from cancer cells, the circulating tumor DNA (ctDNA), which represents from 0.01% to 90% of total cfDNA [33,34,35,36]. Indeed, it is assumed that ctDNA is released in the plasma when superficial tumor cells undergo lysis spontaneously or in response to chemotherapy [37]. The concentration of ctDNA is determined by the presence and size of the cancer [38] and its metabolism and diffusion (clearance, degradation, lymphatic circulation, and other blood processing) [38,39]. DNA fragments released from cancer cells are composed of kilobases from 0.18 to 21 [40,41]. ctDNA can be isolated, amplified with polymerase chain reaction, and then analyzed since it carries the same alterations as cancer: mutations [42], LOH [43], translocations, copy number alterations [44], chromosomal instability [45], and methylations [46]. The identification of ctDNA is possible through the detection of tumor-specific mutations, thus, a previous thorough knowledge of the tumor is mandatory. The potential applications of ctDNA in OC range from the screening to the prediction and monitoring of response to treatment [47]. In approaching PtR recurrent ovarian cancer (ROC) patients, the advantages of analyzing ctDNA are related to the possibility of detecting cancer genetic alterations, which correlate to chemo-resistance and prognosis, and its quantification could be a useful tool.

Some authors evaluated baseline plasma levels of cfDNA in patients with multi-resistant EOC. They found that patients with high cfDNA had a poor outcome relative to lower cfDNA ones after treatment with bevacizumab (PFS 2.9 months vs. 4.2 months, HR 1.98, *p* = 0.002 and OS, 5.0 months vs. 8.1 months, HR 1.66, *p* = 0.02) [48]. Similarly, the measurement of cfDNA is also a promising tool in the future for monitoring the treatment efficacy of PtR OC [49]. Genetic or epigenetic alterations encountered in ctDNA are each discussed in their own sections.

Nevertheless, in the current clinical practice, the utility of ctDNA is largely hampered by its high fragmentation, short half-life (from 15 min to a few hours) [50], and low quantity in the bloodstream [51,52], and the major obstacle is isolating ctDNA from other circulating DNA in the blood sample. Moreover, it could be difficult to determine whether the levels of ctDNA are due to highly proliferating tumors and tumor shedding or to the response to therapies and tumor killing. In addition, the timing at which the samples are collected could be crucial. Finally, the analysis of ctDNA does not allow for the study of other compounds such as RNA, proteins, and metabolites.

#### 3.2.2. Circulating Tumor Cells

CTCs are clones of primary tumor cells released in the bloodstream [53] and are extremely rare in the healthy general population [54]. In particular, they are found in various carcinomas, especially in metastatic ones, and some of these CTCs may be able to colonize distant sites. Only tumor cells with specific features can survive under the stresses of the bloodstream (i.e., flow, immune cells) [55], resist anoikis [56], and have the ability to facilitate metastases development (plasticity, migration, and invasion) through epithelial-to-mesenchymal transition EMT [57,58]. It was proved that neutrophils, platelets, macrophages, and chemokine were involved in CTCs’ protection in this environment [59,60,61,62]. A useful application of CTCs’ isolation in OC is the opportunity to predict platinum resistance and the prognosis and detect mutations related to (MRP1-10, MDR1, ERCC1, RRM1, RRM2) [63,64] through qualitative and quantitative analysis. Levels of CTCs were supposed to correlate with therapeutic response and survival [65], but all in all, data were not consistent (Table 3) [66,67,68,69].

These contrasting pieces of evidence must be considered in the future perspective of using CTCs as biomarkers of OC. Moreover, isolation and characterization of these cells in clinical practice are also extremely obstructed by their scarcity in the bloodstream (from one in 100 million to one in a billion normal blood cells) [70,71,72]. In addition to this, it should be remarked that CTCs half-life is short (about 4 h) after blood draw [73]. At the same time, analysis of CTCs allows the identification of numerous other mutations in a single cell (with respect to cfDNA), leading to a better comprehension of tumor heterogeneity itself [74]. This is an essential advantage of liquid biopsy, particularly CTCs, given the assumption that cancer is not a static disease but a dynamic and mutable process from diagnosis to recurrence. Growing evidence and developments in single-cell isolation techniques, single-cell -omics, and bioinformatics suggest that CTCs display the same heterogeneity as the primary tumor [75]. Indeed, CTCs show an intermediate phenotype between epithelial and mesenchymal and have a highly plastic stem-like state. In the future, the CTC characterization could shed more light on tumor heterogeneity and therapeutic resistance mechanisms and uncover novel therapeutic targets [76].

#### 3.2.3. Extracellular Vesicles

Extracellular vesicles (EVs) include exosomes, microvesicles, and other membranous structures that contain proteins, miRNA, DNA fragments, non-coding RNAs, and lipids. They are abundantly released into the extracellular space by cancer cells and can be easily isolated from various body fluids [77]. Several reports have demonstrated that exosomes can be detected in the bloodstream and ascites of OC patients [78,79]. Compared to CTCs and ctDNA, EVs have the advantage of being more abundant, stable, and accessible [79]. In fact, exosomes can be used as biomarkers for the early diagnosis of cancer and follow-up monitoring. Furthermore, exosomes and their cargoes were found to play a crucial role in disease progression and potentially facilitate chemo-resistance in OC, influencing prognosis. As a result, soluble E-cadherin is highly expressed in the ascitic fluid of women with OC, released in the form of exosomes. It is a potent inducer of angiogenesis and results in a poor prognosis [80]. E-cadherin could be a future therapeutic target, given the availability of E-cadherin (human) monoclonal antibodies. However, the literature evidence on this treatment is still missing. In addition, Peng et al. suggested that exosomes play a role in influencing the immune system; these vesicles, containing heat shock proteins, major histocompatibility complex class I molecule (MHC-I), and Cluster of Differentiation 81 (CD81), could compromise the cytotoxic activity of peripheral blood mononuclear cells, in the presence of dendritic cells [81]. Finally, EVs might be useful in assessing responses to therapy in OC patients. The evidence showed that patients with a good response to treatment develop substantial changes in their level of exosomes (with TGF-β1 and MAGE3/6) after chemotherapy in comparison to patients who did not respond [82]. Although these results show that EVs can play a role in approaching OC patients, the need for fully validated tests (such as Flow Cytometry, Nanoparticle Tracking Analysis, or Electron Microscopy) represents the major limit to their application in clinical practice [83]. Most data related to exosomes and OC derive from exosomal miRNAs and are discussed in the relevant sections.

### 3.3. Epigenetic and Genetic Biomarkers

Overall, cancer development and progression are the results of the accumulation of genetic and epigenetic alterations [84].

#### 3.3.1. Epigenetic Alteration Markers

Epigenetics refers to the alteration of gene expression without any modification to the DNA sequence of the gene itself. This phenomenon is involved in cancer initiation, progression, and eventual resolution. Specifically, epigenetic modifications refer to post-transcriptional gene regulation by miRNAs, DNA methylation, and histone post-translational modifications.

##### MicroRNAs

MiRNAs are short (18–25 nucleotides) non-coding fragments of RNA that bind to and inhibit mRNAs (messenger RNAs). They play a role in cancer development and progression. They can regulate gene expression post-transcriptionally and function as oncogenes or tumor-suppressor genes. Moreover, miRNAs can down-regulate multiple mRNAs and subsequent proteins that are pivotal for drug response, causing platinum resistance. Therefore, inhibiting specific miRNAs may lead to overcoming this condition [85,86]. In addition, miRNAs may also be used as biomarkers for predicting the response to chemotherapy, with the aim of enhancing therapeutic effect and reducing treatment toxicity [87,88,89].

In epithelial OC patients, circulating miRNAs were detected in the serum/plasma (miR-21, miR-141, miR-200a, miR-200b, miR-200c, miR-203, miR-205, miR-214) [90] and in a variety of body fluids, such as urine (miR-30-5p) [91] and ascites (mIR-21, miR-23b, miR-29a) [92]. miRNAs seem to display remarkable stability, despite the presence of RNase in circulation. Indeed, they are protected by membrane-enclosed vesicles such as exosomes and microvesicles, or bound to a carrier protein or lipids (i.e., Argonaute2 and High-Density Lipoprotein) [93,94,95,96].

Several circulating miRNAs are thought to provide useful information for an early diagnosis of OC [97,98,99,100,101,102]. They have also been associated with the prognosis of EOC [103,104,105] and may predict therapeutic response. Actually, despite several studies focused on the effects of tissue miRNAs in modulating the OC cell’s sensitivity to chemotherapeutic agents (e.g., cisplatin, paclitaxel), data relating to circulating ones are scarce, particularly in PtR OC. The most relevant evidence about the potential use of miRNA in approaching PtR OC is summarized in Table 4 [106,107].

All in all, circulating miRNA in PtR OC showed a high predictive value, but data are too limited. Moreover, another drawback is the high cost and limited availability of the test that allows the measurement of miRNA concentration in serum/plasma.

##### DNA Methylation

DNA methylation catalyzed by DNA methyltransferases (DNMT) regulates the genes’ expression, transferring methyl-groups (CH3-) from the S-adenosylmethionine (SAM, methyl donor) to the nucleotide cytosine followed by a guanine (the so-called CpG site) [108]. Aberrant DNA methylation (in excess or in default) leads to chromosome instability and changes in gene expression and correlates to the development and progression of cancer. For 20 years, it has been evident that cancer-related DNA methylations are chemically and biologically stable in blood and can be detected in the serum/plasma of patients. Additionally, DNA methylation analysis has the advantage of not requiring a scan of the whole gene but can rather be focused directly on the CpG sites [108,109,110]. It has been shown that several tumor suppressor genes involved in OC are hyper- or hypomethylated. This phenomenon was observed in all pathological grades and stages [110]. As the methylation can be detected in a blood test at the time of primary diagnosis or relapse, it could possibly give information about a response to platinum-based medication and prognosis. Evidence about the value of methylation in OC are summarized in Table 5 [109,110,111,112,113,114].

Some of the most relevant genes involved in platinum-resistance in OC include Phosphatase and Tensin Homolog (PTEN), Regulator of G Protein Signaling 2 (RGS2), Family with Sequence Similarity 83 member A (FAM83A), Myosin XVIIIB (MYO18B), which are hypermetylation [115,116] and Msh Homeobox 1 (MSX1) and Transmembrane Protein 88 gene (TMEM88) with hypomethylation [117,118].

Even if most of these data come from newly diagnosed OC or in vitro, it is reasonable to assume that the methylation of cfDNA in blood could also serve as a useful marker in PtR OC. Finally, methylation profiles could also be a target for testing new combination treatment regimes. Preclinical evidence from different cancers (including OC) showed that hypomethylating agents can re-sensitize cancer cells to platinum in vitro and in murine models by restoring tumor-suppressor genes expression (such as RASSF1A, BRCA1, DAPK, OPCML, and hSulf-1) [119,120,121,122].

Hence, several studies have assessed the role of Hypomethylating agents (HMAs) in PtR OC. First of all, the administration of decitabine in combination with carboplatin was tested, but the results were contrasting [123,124,125,126]. Other authors assessed the role of azacitidine, a hypo-methylating agent, in combination with carboplatin for platinum-resistant HGSOC [127] in a phase Ib-IIa study. In 30 enrolled patients, an Overall Response rate (ORR) of 13.8% was found (4/29; 95% CI, 10.1–17.5%): 1 clinical complete response (CR), 3 clinical partial responses (PRs) and 10 stable diseases. The PFS was 5.6 months and median OS 23 months. Moreover, azacitidine seems to enhance the sensitivity to platinum in association with a DR4-mediated caspase 8-dependent apoptosis13. Therefore, a correlative analysis showed that DR4 methylation in peripheral blood leukocytes decreased during treatment in 75% of objective responders (3/4), more than in non-responders (5/13, 38%) [127]. Most recently, a phase II randomized trial compared the combination of guadecitabine and carboplatin (51 patients) versus treatment of choice (TC topotecan, pegylated liposomal doxorubicin, paclitaxel, or gemcitabine) (49 patients, of which 27 crossed over to the other arm) in PtR OC. This trial did not show superiority for PFS of the combination versus TC (16.3 weeks vs. 9.1 weeks *p* 0.07), while the 6-month PFS increased (37% vs. 11%, P 0.003) [128]. In summary, identifying DNA methylation in the blood of patients may guide the physician in predicting platinum resistance and, in some cases, permit restoration of the sensitivity to this agent. Thus, it is a promising area but nowadays limited in clinical practice.

##### Histone Modifications and Involved Enzymes

Histones are small basic proteins bound to DNA in eukaryotic cells. Their principal function is to regulate gene expression and DNA packaging around nucleosomes, the functional units of chromatin. The presence of histones in the bloodstream is a result of tumor cell death (apoptosis and necrosis) or active release from living cells. Consequently, circulating histones reflect changes in tumor cells, and, therefore, are promising non-invasive biomarkers in several cancers [129]. In particular, an increasing level of circulating nucleosomes/histones has recently been identified in the blood of oncologic patients [130,131], and a quantitative measurement can be useful in predicting tumor responses to chemotherapeutic agents in various cancer types [132]. Despite the fact that these data do not refer to OC, it is likely that the high level of circulating histones has the same correlation with the diagnosis and prognosis of this disease. Moreover, histones could undergo modifications by enzymes. These post-translational modifications of histones included phosphorylation, acetylation, methylation ubiquitylation, glycosylation, SUMOylation, ADP (adenosine diphosphate)-ribosylation, and carbonylation [133] and were proved to be correlated to cancer development and its prognosis [134,135,136]. Particularly in OC, the importance of the detection of histone in fluids is attributable to the fact that histone-modifying enzymes have recently been studied as a possible targeted treatment for this disease, especially Histone Deacetylases (HDACs). Normally, histone acetyltransferases catalyze the transfer of an acetyl functional group from a donor (e.g., Acetyl CoEnzyme A) to a lysine residue protruding from the histone of the nucleosome. The acetylation causes the loss of positive charge on histones and weakens the bonds of DNA components (relaxed structure of chromatin). This euchromatin is more accessible to gene transcription enzymes. Conversely, deacetylation by HDACs leads to the formation of a more condensed DNA (heterochromatin), not transcriptionally active. Among HDACs, sirtuins (SIRT) regulate cell cycle progression, apoptosis, cell senescence, and oxidative stress resistance, leading to tumorigenesis [137,138]. Given the evidence of a link between SIRT 1 and stemness (cancer stem cells), SIRT1 is considered to be associated with recurrence and drug resistance. Indeed, SIRT1 was proved to significantly enhance the proliferation (*p* < 0.05), chemo-resistance (*p* < 0.05), and aggressiveness of OC cells [139]. Thus, research on SIRT1 is important for developing novel treatment strategies as an adjuvant to conventional therapies to overcome drug resistance [140]. Among current HDAC inhibitors, suberoylanilide hydroxamic acid, valproic acid, and romidepsin have been tested in ovarian cancer as single agents or in combination with other drugs. The use of therapy targeting modified histones and the enzymes regulating them is quite promising in ovarian cancer [141,142,143,144]. However, nowadays, robust clinical trials are unavailable, making it difficult to ascertain whether this treatment offers beneficial clinical outcomes with tolerated side-effect profiles. Moreover, whether these drugs will be more efficacious as single agents or in combination remains to be determined. Consequently, it is still premature to argue that histone modifications can be used as circulating biomarkers in OC and further evidences are necessary.

#### 3.3.2. Genetic Alteration Markers

##### TP53 Mutations

The p53 is a nuclear protein that acts as a transcriptional regulator involved in multiple cellular processes. This protein is encoded by the tumor suppressor gene TP53, located on chromosome 17 [145]. p53 can activate DNA repair proteins when the DNA has sustained damage: indeed, p53 leads to the arrest of cell growth by holding the cell cycle at the G1/S transition. In this way, DNA repair is allowed, and cell death occurs if DNA damage is irreparable. Given its essential role, p53 is frequently mutated in cancer. Regarding OC, pathogenic TP53 mutations have been identified in more than 99% of HGSOC cases [146,147], and approximately 80% of them are missense mutations, in which a single nucleotide is substituted by another [148]. Most of these mutations result in loss of p53 suppressive activities (loss-of-function) [149]. Nevertheless, mutant p53 proteins were additionally proved to be able to gain oncogenic functions that provide cells with growth and survival abilities (gain-of-function) [150,151]. The presence of TP53 mutations can be detected by finding anti-p53 antibodies in the bloodstream due to the humoral response associated mainly with missense mutations and accumulation of mutant protein in the tumor [152,153]. The immunoglobulins are supposed to be a useful marker at the diagnosis of ovarian cancer [154], while their prognostic significance is still unclear [155,156,157]. Moreover, TP53 mutations can be detected in ctDNA from patients with advanced HGSOC. Some data underlined the diagnostic value of TP53 mutations in serum ctDNA that can be detected at baseline, which are not present in cfDNA after chemotherapy, and which re-appeared at the development of relapse [158]. Regarding other functions, a retrospective analysis demonstrated that TP53 mutations in ctDNA correlate to prognosis (time-to-progression TTP) and play a role in monitoring the response to chemotherapy with more efficacy than CA125. As a matter of fact, in recurrent disease, TTP was significantly longer in cases of low pre-treatment levels of TP53 mutant allele fraction (below the median level) as opposed than high levels (above the median) (*p* = 0.001, 168 vs. 245 days, HR 0.33 95% CI 0.17–0.64). Moreover, a decrease in TP53MAF of >60% after the first cycle of chemotherapy was proven to be an independent predictor of TTP in multivariable analysis (HR 0.22, 95% CI 0.07–0.67, *p* = 0.008), while a decrease <60% was associated with poor response and worse TTP (median TTP 76 days vs. 229 days, *p* = 0.001, HR 0.08, 95% CI 0.02–0.34) [159].

##### Homologous Recombination and BRCA Genes

Homologous recombination is responsible for the repair of DNA double-strand breaks that occur in case of damaging insults (such as ionizing radiation and chemotherapy) [160]. In the last decade, it has been demonstrated that approximately 50% of HGSOCs have a homologous recombination deficiency (HRD) [161], caused by the mutation of several genes, especially Breast Cancer susceptibility gene 1 and gene 2 (BRCA 1 and BRCA 2) mutation (somatic or germinal). The frequency and modalities of detection of these alterations in HGSOC are summarized in Figure 4 [161].

HRD and BRCA status is fundamental for patient framing and counseling. First of all, HRD and BRCA mutation could predict prognosis (prognostic value). Several studies reported a longer OS and PFS in BRCA positive patients as opposed to non-carriers [162,163,164,165,166], probably due to a higher platinum sensitivity. In fact, preclinical evidence showed that the deficiency of a specific DNA repair pathway (especially HRD itself) was associated with a higher sensitivity to platinum drugs, precisely because the main target of platinum compounds is DNA [167,168,169,170,171,172]. Thus, in recurrent OC with PFI < 6 months, more patients with BRCA mutations were proven to have a response to re-treatment with platinum-based chemotherapy in comparison to wild-type ones (80% vs. 43.6%). The same occurred in case of non-platinum regimen (42.8% vs. 16.1%, *p* = 0.001) [173]. Finally, the BRCA status might be a predictor of response to other agents, such as Poly (ADP-ribose) polymerase inhibitor (PARPis) (predictive role). Indeed, PARPis prevent the mechanism of single-strand DNA repair and lead to synthetic lethality in HRD or BRCA carriers. More recent evidence shows that the detection of mutations has been considered the only possibility for targeted maintenance therapy with PARPis after a response to platinum therapy in newly diagnosed or recurrence settings. However, nowadays, it is clear that PARPis are also active in wild-type populations, especially if HRD is positive [174,175,176]. However, in the USA, the germline BRCA mutation is still needed for the administration of olaparib in monotherapy in recurrent ovarian cancer, regardless of platinum sensitivity, based on results of Study 42 [177]. As a result, currently, the BRCA test (germinal on the bloodstream and/or somatic on tissue sample) is part of clinical practice. The BRCA status is surely an easily available prognostic and predictive biomarker, also in PtR OC.

Despite the platinum sensitivity associated with BRCA mutation, reversion mutations in tumor cells (somatic base substitutions or insertions/deletions) that restore the open reading frame (ORF) of the primary germline BRCA1 or BRCA2 mutation can occur, resulting in a functional protein and a proficient homologous recombination DNA repair [178,179,180]. Hence, knowledge of the presence of these alterations in cancer cells is a very useful tool for the identification of patients with BRCA mutation who will not respond to platinum, avoiding unsuccessful treatment.

Even if tumor biopsy is currently the only way to detect somatic mutation, the analysis of cfDNA could be a future winning strategy to obtain this information through a blood sample [181,182,183]. Data of BRCA reversion mutations in cfDNA from patients with other tumors are present in literature [184]. Regarding OC, some authors demonstrated the possibility of detection of reversion mutations in BRCA mutated PtR ROC, with a high concordance to tissue samples (79%) [49,185]. Moreover, BRCA reversion mutations were identified in cfDNA particularly in platinum-refractory and –resistant patients, compared with platinum-sensitive ones (18% and 13% vs. 2%, respectively, *p* = 0.049) [186].

Currently, BRCA 1 and BRCA 2 mutations are searched in OC patients in clinical practice. However, the development and the spread of tests that determine HRD status can provide further information and permit a more personalized approach.

### 3.4. Angiogenic Biomarkers

Angiogenesis is a process characterized by the generation of new blood vessels from pre-existing ones and plays a role in the development, growth, and metastatic spread of solid cancers. The angiogenesis is regulated by multiple mechanisms involving growth factors. Among them, the most studied one is the VEGF, which plays an essential role in many tumor types [187,188,189,190]. VEGF is secreted by cancer cells, especially in case of hypoxia and scarcity of nutrients [191,192]. VEGF promotes angiogenesis binding to its tyrosine kinase receptors (VEGFR) expressed in endothelial cells. A meta-analysis of data regarding newly diagnosed OC revealed that high levels of soluble VEGF (sVEGF) identified a subgroup of patients with a higher risk of death and/or recurrence since, at multivariate analysis, sVEGF was proved to be an independent prognostic factor for OS and PFS [193]. However, this predictive potential of serum levels of sVEGF was not confirmed in ROC [194]. In addition, the role of VEGF in predicting response to platinum first-line therapy was investigated. No differences in sVEGF levels were detected in patients with platinum-sensitive and -resistant disease at baseline (*p* = 0.058) and during upfront treatment (at third and sixth cycle, *p* = 0.09). Moreover, in this population, haplotypes were also studied, and the multivariate analysis showed that PFS in the case of AGCGC haplotype was significantly improved compared to patients with other ones (HR 1.9, *p* = 0.036). However, no significant associations were found between haplotypes and platinum resistance (*p* = 0.30) [195]. In contrast, in recurrent platinum-resistant OC, it was found that a rapid decrease in serum VEGF-A levels (>50%) after treatment with bevacizumab and gemcitabine was associated with worse RR (0% vs.75%, *p* < 0.01), clinical benefit (60% vs.100%, *p* = 0.02) and survival (PFS 7 vs. 10 months, *p* < 0.01; OS 17 vs. 26 months, *p* = 0.04). Moreover, the median serum VEGF-A level before the first cycle was higher in the group with a rapid decrease of VEGF-A (61.2 vs. 3.7 pg/mL, *p* < 0.01) [196].

Moreover, other authors tested the efficacy of bevacizumab in multi-resistant disease, and, at the same time, levels of VEGF were assessed prior to each cycle. On the whole, the results of drug activity were positive (the overall response rate was 30% according to CA 125. Median PFS 5.9 months (95% CI, 3.5–9.4), median OS 8.6 months (95% CI, 6.6–12.8). Baseline high levels of VEGF (above the median) appeared to be predictive of no response to bevacizumab (responders were 60% of patients with low VEGF and 0% of those with high-level *p* = 0.0007). In accordance, higher VEGF levels resulted in a worse PFS and OS in respect to VEGF below the median (PFS 3.5 months vs. 10 months, *p* = 0.047; OS 5.7 months vs. not reached, 1-year OS 22% vs. 68%, *p* = 0.01). Furthermore, in this setting of patients, VEGF and VEGFR1 gene polymorphisms did not reveal any association with response rate or survival [197].

### 3.5. Immune-Related Biomarkers

Cancer-associated inflammation plays a determinant role in tumor-initiating, proliferation, and survival of malignant cells. Moreover, its correlation with the outcome of patients affected by different types of malignancies, including ovarian cancer, has recently been observed.

Cytokine and chemokine signaling pathways are involved in OC progression and in response to chemotherapy. Circulating cytokines and several other inflammatory biomarkers have been investigated as prognostic factors in OC patients. Most relevant evidence regarding the value of the neutrophil-lymphocyte ratio (NLR) and platelet-lymphocyte ratio (PLR) with a possible role in PtR OC patients is summarized in Table 6 [198,199,200].

Even if not all these data referred to PtR patients, it is conceivable that a possible prognostic [198] and predictive value [199,200] of PLR and NLR is also applicable in this setting of disease. Fibrinogen also seems to be able to predict the response to chemotherapy. Data from a retrospective study has shown that high plasma fibrinogen levels, combined with NLR, could be predictive of platinum resistance (*p* = 0.02) and shortened PFS (*p* = 0.02) in OC patients [201]. These inflammatory biomarkers, such as NLR and PLR, would be able to predict prognosis and chemotherapeutic efficacy due to their ability to reflect systemic inflammation and organ dysfunction. High NLR levels might indirectly indicate poor lymphocyte-mediated immune response against cancer. Moreover, neutrophils seem to accelerate tumor progression by transforming growth factor β (TGF-β) pathways. On the other hand, platelets produce various types of cytokines, including vascular endothelial growth factor (VEGF), an important factor for tumor angiogenesis. All these factors are involved in poor prognoses.

The assessment of these parameters is simple and economically viable. However, to date, an important limit arises in the form of the cut-off value of these biomarkers, which has not been universally established but is instead chosen arbitrarily. These differences in cut-off values cause difficulties in terms of using them in clinical practice. The involvement of the immune system in cancer development is also the foundation of the success of immunotherapy in treating some neoplasms. Actually, in OC the role of immunotherapy is still uncertain, and results from immunotherapeutic agents administered alone are not as positive as expected, and nowadays, the use of immunotherapeutic drugs in OC is limited to clinical trials. Since few and ineffective chemotherapeutic agents are currently available for PtR OC, immunotherapeutic drugs, especially in combination with other agents [202,203,204,205,206,207], could represent a future successful strategy and change the prognosis of affected women. Consequently, the main challenge regarding the identification of populations that could benefit more from this approach and biomarkers that predict the response to this treatment deserves a separate discussion.

Currently, available immunotherapy agents are monoclonal antibodies targeting programmed cell death protein (PD-1), programmed cell death protein ligand 1 (PD-L1), and cytotoxic T-lymphocyte antigen 4 (CTLA-4), which act as immune checkpoint inhibitors (ICIs) [208,209]. Among biomarkers predicting response to ICIs, the main ones are intratumor PD-L1/PD-1/CTLA-4 expression, density of TILs, tumor mismatch-repair (MMR) deficiency. In clinical practice, most of them are obtained from tumor samples. Some authors evaluated the function of circulating T-cell [210,211] and B-cells [212] in OC, showing a correlation between the activation of the immunologic system and response to chemotherapy and vice-versa. These results led the same authors to sustain that the response to chemotherapy, and resulting high levels of circulating lymphocytes, may provide an opportunity for the success of subsequent immunotherapy. In addition, PD-1 and PD-L1 also have soluble forms (sPD-1 and sPD-L1) in the serum. Their levels seem to correlate with response to immunotherapy and survival in several types of malignancies [213], but data are conflicting. Interestingly, despite HRD/BRCA-mutated OCs displaying higher levels of genetic instability, potentially resulting in higher immunogenicity, HRD and BRCA mutations failed to be associated with a better response to ICIs, while the fraction of genome altered (FGA) should be investigated further as a biomarker of response to immunotherapy in OC [214].

Some studies suggest that low levels of sPD-L1 may correlate with longer survival in patients with non -small cell lung cancer, multiple myeloma, renal cell carcinoma [213]. Conversely, it has been reported that in melanoma patients treated with ICIs an increase in sPD-L1 was associated with PRs [215]. To date, the reasons for this dual effect remain unknown. Other blood parameters examined to predict the response to immunotherapy in malignancies are serum lactate dehydrogenase (LDH), NLR, absolute neutrophil counts (ANC), absolute lymphocyte counts (ALC), absolute monocyte counts (AMC), absolute eosinophil count (AEC). However, also in these cases, data in OC is scarce.

Regarding new immunotherapeutic strategies that differ from ICIs, some authors evaluated the role of immune system status in patients treated with abagovomab (high affinity murine monoclonal antibody specific for CA125) after CR to primary surgery and platinum- and taxane-based chemotherapy. They found that higher levels of IFN-γ producing CD8+T cells were associated with a better Relapse Free Survival (RFS) than those with fewer IFN-γ producing CD8+T cells (*p* < 0.05) [216]. Moreover, it was demonstrated that the efficacy of oregovomab (another anti CA125 antibody) correlated to a less suppressive immune environment before treatment and a low number of circulating myeloid-derived suppressor cells, subset type 4 (MDSC4), and low neutrophil-and-monocyte to lymphocyte ratio (NMLR) were significantly associated to RFS (MDSC 4: *p* = 0.012, NMLR *p* = 0.0014). NMLR was related also with OS (*p* = 0.048) [217]. Although these data do not come from PtR OC, the same results can be expected in this patient setting.

## 4. Conclusions and Future Directions

Circulating biomarkers could help gynecologist oncologists deal with recurrent OC after a short PFI, and their investigation is a promising and growing field. In clinical practice, most of the information can be obtained from the immunohistochemical study of the tissue; however, circulating biomarkers have the advantage of a non-invasive collection, thus being easily executable and repeatable.

Currently, the biomarkers routinely used in clinical practice in OC patients are CA125, HE4, and BRCA/HRD assessment. However, these are not completely satisfying in guiding clinical management, and greater efforts are needed to provide new useful tools. Potential circulating biomarkers addressed in this review and their value are summarized in Table 7.

Certainly, the most relevant value is the prediction of the treatment efficacy, as these biomarkers could potentially predict the response to platinum and other agents. Moreover, several biomarkers represent targets for available drugs and thus could identify patients who benefit most from a personalized approach. Indeed, using circulating biomarkers and liquid biopsy in PtR OC ideally allows assessing the instantaneous molecular, genetic and epigenetic profile of cancer cells selected as platinum-resistant clones from previous therapies. This information is fundamental in the precision medicine era, in which new biological and targeted therapies are being discovered continuously, especially in case of limited treatment options such as for PtR OC patients. Finally, circulating biomarkers could be non-invasive tools able to evaluate the response to ongoing treatments, thus rapidly guiding the medical choice for eventual further chemotherapy cycles or the need to change treatment strategy.

Regarding the prognostic role, some biomarkers aid in counseling patients due to their association with a major risk of recurrence or death, regardless of treatment response. However, an exhaustive prognostic biomarker has not yet been identified. The prognosis is the result of several factors, depending on the biology of cancer (histotype and grading of differentiation), disease spread, patient characteristics (performance status, age, and comorbidity), and treatment received (optimal surgical cytoreduction, response to chemotherapy). Besides, as biomarkers can also complement each other, using a single biomarker to predict prognosis could be highly limiting. Furthermore, the lack of standardized detection methods, the scarce accessibility in the territory, the high costs, and the uncertainty about cut-off levels could hinder the use of circulating biomarkers in routine clinical practice.

In conclusion, in the future, circulating biomarkers could represent the verge of a breakthrough in approaching PtR OC, influencing treatment decisions due to their characteristic of detecting the heterogeneity of this disease in different phases. However, the validation of circulating biomarkers is challenging, and further studies are required to overcome their limits. Translational analysis of wide clinical trials and prospective studies will pave the way for promoting implementation in clinical routines.

## Figures and Tables

**Figure 1 ijms-22-13650-f001:**
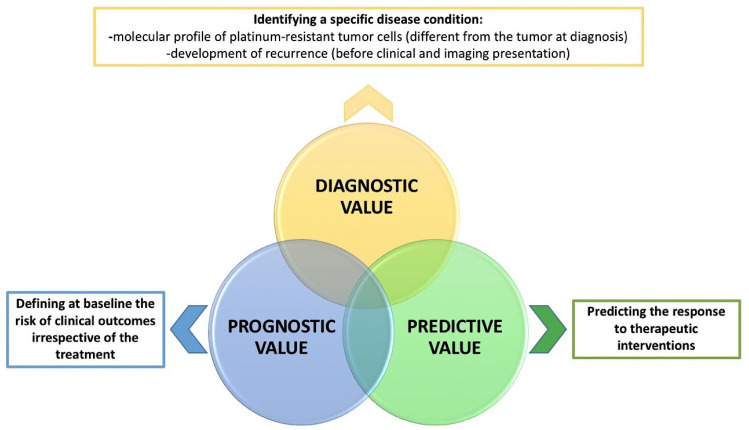
Potential values of biomarkers in PtR OC.

**Figure 2 ijms-22-13650-f002:**
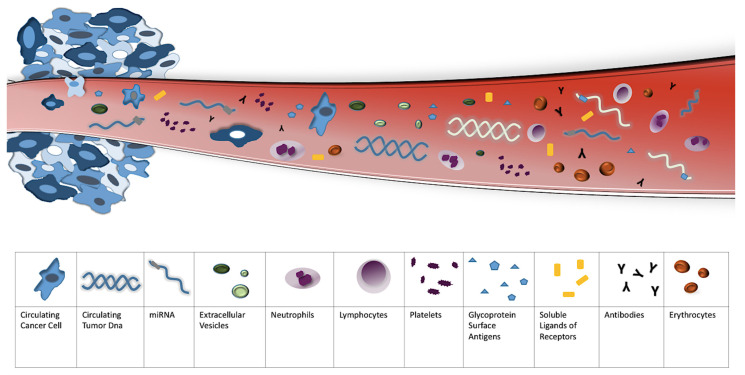
Circulating biomarkers in PtR OC.

**Figure 3 ijms-22-13650-f003:**
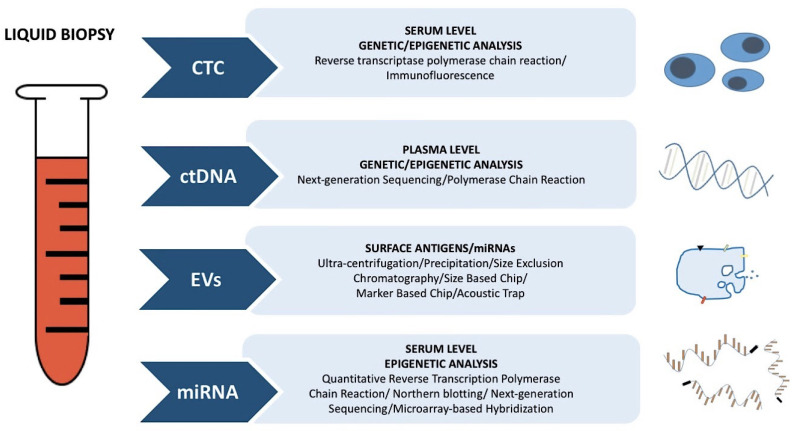
Liquid Biopsy: what is assessed and methods of detection.

**Figure 4 ijms-22-13650-f004:**
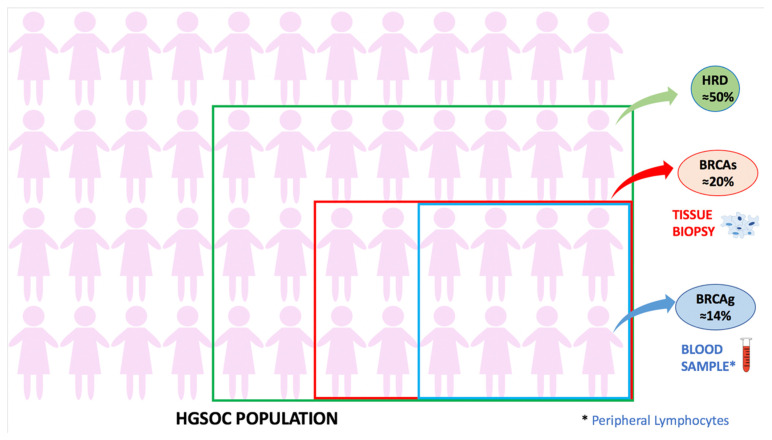
HRD and BRCA1/2 mutation in HGSOC. Approximately 50% of HGSOCs have homologous recombination deficiency (HRD). Among genes, BRCA1/2 are the most common involved: about 14% of HGSOC patients were reported to have BRCA1/2 mutation detectable in a blood sample (germinal mutation, analysis of peripheral Lymphocytes). While a BRCA1/2 mutation found only in a tissue sample is present in about 20% of the affected women (somatic mutation). Other genes with a role in HRD are CDK12 (3% of cases), RAD51C (2%), EMSY (6%), and PTEN (7%).

**Table 1 ijms-22-13650-t001:** Characteristics of circulating biomarkers and tumor biopsy.

Circulating Biomarkers and Liquid Biopsy	Tumor Biopsy
Material derived from cancer detectable in bloodstream, urine, or peritoneal fluid	Material obtained from a sampling of tissue lesion
Non-invasive procedure	High invasive procedure
Real-time follow up	Impracticable for real-time follow up
Quick and easily repeatable procedure for obtaining the samples	Difficult to repeat and depend on the correctness of the procedure
No surgical complication or pain	Risk of surgical complication and pain
Lack of well-defined practice rules and standardizing protocols	Clinically validated and standard for histologic diagnosis
Less cost (with some exceptions)	High cost
Assessment of tumor heterogeneity in different phases of the disease	Failure to reflect tumor heterogeneity
Low concentrations and easily degradable material	Higher concentration and fixed material
Less specificity	Higher specificity
Specialized laboratory	Histology laboratory

**Table 2 ijms-22-13650-t002:** Discussed circulating biomarkers in OC.

Type of Circulating Biomarker	
**Glycoprotein Biomarkers**	CA 125
	HE4
	Mesothelin
**Liquid Biopsy**	ctDNA
	CTCs
	EVs
**Epigenetic and Genetic Markers**	miRNA
	DNA methylation
	Histone modification
	TP53 mutation
	HRD-BRCA1/2 mutation
**Immune-Related Biomarkers**	NLR
	PLR
	Circulating T-cell
	Circulating B-cell
	sPD-1/sPD-L1
	MDSC4
	NMLR
**Angiogenic Markers**	sVEGF

Abbreviations: BRCA Breast Cancer susceptibility gene, CTCs Circulating Tumor Cells, ctDNA Circulating Tumor DNA, EVs Extracellular Vesicles, HE Human Epididymis Protein 4, HRD Homologous Recombination Deficiency, MDSC4 Circulating Myeloid-Derived Suppressor Cells type 4, miRNAs Micro RNAs, NLR Neutrophil-Lymphocyte ratio, NMLR Neutrophil-and-Monocyte to Lymphocyte Ratio, PLR Platelet-Lymphocyte Ratio, sPD-1 soluble form of Programmed Cell Death Protein 1, sPD-L1 soluble form of Programmed cell Death Protein Ligand 1, sVEGF soluble form of Vascular Endothelial Growth Factor.

**Table 3 ijms-22-13650-t003:** Most relevant evidence about the role of CTCs in PtR OC.

Author, Year	Material and Methods	Results	Conclusions
Kuhlmann JD. 2014 [66]	143 new diagnosed EOC pts. Immunomagnetic CTCs enrichment targeting EPCAM and mucin 1 followed by multiplex reverse transcription PCR. Classified according to the presence of CTCs expressing ERCC1 (ERCC1^+^ CTCs vs. ERCC1^−^CTCs).	**Platinum resistance ERCC1^+^ CTCs vs. ERCC1^−^CTC** OR, 8.5 (1.7–43.6), *p* = 0.01	The presence of CTCs expressing ERCC1 is an independent predictor of platinum resistance
Obermayr E. 2013 [67]	216 pts with EOC. RT-qPCR analysis of EpCAM in CTCs at follow up.	**Frequency of CTCs with overexpression of PPIC gene in PtR vs. platinum sensible patients at follow up:** 35.7% vs. 10.1%, *p* = 0.024	CTCs with overexpression of PPIC gene correlate with platinum resistance
Poveda A. 2011 [68]	216 pts ROC (PtR 34%) treated with PLD ± trabectedine CTCs isolated from blood using Cell Search system and reagents (Veridex) Classified according to CTCs at baseline: CTCs ≥ 2 vs. CTCs < 2.	-**PFS ≥2 CTCs vs. <2 CTCs:** 3.2 months vs. 6.6 months; *p* = 0.0024. -**OS ≥ 2 CTCs vs. <2 CTCs:** 12.4 months vs. 20.6 months; *p* = 0.0017. -**Multivariate analysis:** PFS HR 1.58 (0.99–2.53) *p* = 0.058 -**Multivariate analysis:** OS HR 1.54 (0.93–2.54) *p* = 0.096	Levels of CTCs seem to correlate with platinum resistance and worse survival, but data are inconsistent
Lee M. 2017 [69]	30 pts with ROC (PtR 60%) Fresh peripheral blood samples collected in EDTA vacutainer tubes, engaging of biotin-doped PPy-deposited microfluid device, polydimethylsiloxane microchannels coniugated with streptavidin and exposed to antibodies against EpCAM, TROP2, EGFR, vimentin, and N cadherin. Classified according to CTCs cluster positivity.	-**OS pts with CTCs cluster vs. pts without CTCs cluster:** 21 vs. 74 months, *p* = 0.008. -**Multivariate analysis OS:** HR 1.3 (0.94–17.149) *p* = 0.94 −65.2% of patients with CTCs cluster showed platinum resistance (*p* = 0.001).	Levels of CTCs seem to correlate with platinum resistance and worse survival, but data are inconsistent

Abbreviations: CTCs Circulating Tumor Cells, EDTA Ethylenediaminetetraacetic acids, EGFR Epidermal Growth Factor Receptor, EOC Epithelial Ovarian Cancer, EpCAM Epithelial Cellular Adhesion Molecule, ERCC Excision Repair 1 protein, HR Hazard Ratio, OR Odds Ratio, OS Overall Survival, PFS Progression Free Survival, PLD Pegylated Liposomal Doxorubicin, PPIC Cyclophilin C gene, PtR Platinum resistant, Pts Patients, ROC Recurrent Ovarian Cancer, RT-qPCR Real-Time quantitative Polymerase Chain Reaction, TRP-2 Tyrosinase-related protein 2.

**Table 4 ijms-22-13650-t004:** Most relevant evidence about the potential value of miRNA in PtR OC.

Author, Year	Material and Methods	Results	Conclusions
Benson EA. 2015 [106]	14 pts with PtR ROC Evaluation of plasma miRNAs in predicting the response to carboplatin and decitabine	10 miRNAs changed in concentration at the end of the first cycle of treatment (ranging from 2.9 to 4, *p* < 0.05) and were associated with response. Lower concentrations miR-148b-5p predicted worse PFS (*p* = 0.015).	miRNA analysis predicts the response to chemotherapy and prognosis.
Vigneron N. 2020 [107]	35 pts with ROC (16.9% PtR) Evaluation of miR-622 levels at relapse Classification according to Levels: >0.34 zmol/mL vs. <0.34 zmol/mL	OS miRNA > 0.34 zmol/mL vs. <0.34 zmol/mL: 7.9 months vs. 20.6 months, HR 3.15, *p* = 0.006	miRNA analysis predicts prognosis

Abbreviations: miRNA micro Ribonucleic Acid, OS Overall Survival, PFS Progression-Free Survival, PtR Platinum-Resistant, Pts Patients, ROC Recurrent Ovarian Cancer.

**Table 5 ijms-22-13650-t005:** Most relevant evidence about the value of methylation alteration in OC.

Author, Year	Material and Methods	Results	Conclusions
Losi L. 2018 [111]	102 OC vs. 17 normal ovarian samples Analysis of promoter regions of 41 genes DNA methylation profiling through the MLM	% of hypermethylated promoter genes: In normal ovarian tissues: 29% In serous, endometrioid, and mucinous carcinomas: 32%, 34%, and 45%, respectively.	OC is characterized by a slight increase of hypermethylation
De Caceres II. 2004 [110]	50 pts with new diagnosed OC or PPC (Stage I-IV) vs. 40 healthy women (control group) Specimens/serum/peritoneal fluid Sensitive methylation-specific PCR	% of hypermethylated BRCA 1 and/or RASSF1A: 68% (regardless FIGO stage) vs. 0% in control group.	Promoter hypermethylation is a common and relatively early event in ovarian tumorigenesis
Cacan E. 2016 [112]	Chemo-resistant OC cells vs. chemo-sensitive OC cells Cell-surface staining through primary labeled antibodies: phycoerythrin-conjugated OX-40L, 4-1BBL, PD-L1, MHC-I	The expression of positive co-stimulatory molecules of T cell, OX-40L and 4-1BBL, is suppressed due to DNA hypermethylation and histone deacetylation in chemo-resistant cells compared to parental chemo-sensitive OC cells.	Hypermethylation correlates with chemo-resistance in OC
Gifford G. 2004 [113]	138 OC pts from SCOTROC1 trial (paired samples) Evaluation of methylation of the hMLH1 in plasma before the chemotherapy and at recurrence	Methylation of hMLH1 is increased at relapse 25% (34 of 138) of relapse samples have hMLH1 methylation that is not detected in matched pre-chemotherapy plasma samples hMLH1 methylation in cfDNA at relapse correlated with poor survival: HR 1.83, *p* 0.017 Patients with hMLH1 methylation and PFS < 6 months from last platinum were more in percentage than patients without the epigenetic alteration (45% vs. 39%)	The acquisition of hMLH1 methylation in plasma DNA after chemotherapy predicts poor survival for ovarian cancer patients
Teschendorff AE. 2009 [109]	113 OC pts (vs. 148 healthy controls) 27,000 CpGs screened	2714 cancer-related CpGs were identified 56% of cancer-related CpGs were hypomethylated Amongst the 50 CpGs with the highest correlation to cancer, as much as 87% were hypomethylated.	Hypomethylation is correlated with OC
Liao P. 2014 [114]	168 tissue samples from patients with OC Evaluation of DNA methylation in OTICs through qRT–PCR, quantitative methylation-specific PCR, and pyrosequencing	In case of hypomethylation of ATG4A and HIST1H2BN in OTICs: PFS: HR, 1.8 (1.0–3.6) OS: HR, 1.7 (1.0–3.0) [118]	In OTICs, hypomethylation of ATG4A and HIST1H2BN is associated with poor prognosis

Abbreviations: ATG4A Autophagy Related 4A Cysteine Peptidase gene, BRCA Breast Cancer gene, cfDNA cell-free DNA, DNA Deoxyribonucleic acid, HIST1H2BN Histone H2B type 1-N, hMLH1 MutL homolog 1, HR Hazard Ratio, MHC-I Major Histocompatibility Complex Class I, MLM Methylation Ligation-dependent Macroarray, OC Ovarian Cancer, OS Overall Survival, OTICs ovarian tumor-initiating cells, OX-40L OX40 Ligand, PCR Polymerase Chain Reaction, PD-L1 Programmed cell Death Protein Ligand 1, PFS Progression-Free Survival, PPC Primary Peritoneal Cancer, Pts patients, RASSF1A Ras Association Domain Family 1 Isoform A, RT-qPCR Real-Time quantitative Polymerase Chain Reaction, SCOTROC1 Scottish Randomised Trial in Ovarian Cancer 1, 4-1BBL 4-1BB Ligand.

**Table 6 ijms-22-13650-t006:** Most relevant evidence about the potential role of PLR and NLR in PtR OC.

Author, Year	Material and Methods	Results	Conclusions
Zhu Y. 2018 [198]	2919 pts with OC (meta-analysis) * Correlation between level of PLR and NLR > cut off and survival	**PLR > cut off** - OS: metaHR 2.53 (2.16–2.96) - PFS: metaHR 2.48, (2.10–2.96) **NLR > cut off** - OS: metaHR 2.21 (1.95–2.52) - PFS: metaHR 1.36 (1.17–1.57)	Higher value of PLR and NLR are associated with worse ovarian cancer survival
Miao Y. 2016 [199]	344 pts with OC (28% PtR) * (216 serous OC) Evaluation of NLR and PLR	**Predictive values for platinum resistance:** - PLR > 207: SN 60.42%, SP 85.48%, *p* < 0.001 - NLR > 3.02: SN 75%, SP 81.45%, *p* < 0.001	Assessment of NLR and PLR has potential clinical value in predicting platinum resistance in patients with EOC
Kim HS. 2016 [200]	109 pts with CCOC (18.3% PtR)	PLR ≥ 205.4 predicted non-CR (accuracy, 71.6%) **Predictive values for platinum resistance:** - NLR ≥ 2.8: SN 68.4%, SP 65.1%, *p* < 0.01 - PLR ≥ 178.3: SN 68.4, SP 55.4%, *p* = 0.02	NLR and PLR value correlate with platinum resistance in patients with CCOC

* The meta-analysis also includes 344 OC pts from Miao Y et al., 2016 [199]. Abbreviations: CCOC Clear Cell Ovarian Cancer, CR Complete response, EOC Epithelial Ovarian Cancer, HR Hazard Ratio, NLR Neutrophil-Lymphocyte Ratio, OC Ovarian Cancer, OS Overall Survival, PFS Progression-Free Survival, PLR Platelet-Lymphocyte Ratio, PtR Platinum resistant, Pts Patients, SN Sensitivity, SP Specificity.

**Table 7 ijms-22-13650-t007:** Circulating biomarkers in PtR OC, their potential role and limits.

	Diagnostic Value	Prognostic Value	Predictive Value	Currently Used in Clinical Practice	Limits
**Glicoprotein markers**					Low specificity
**ctDNA**					High fragmentation, low stability, and low quantity in bloodstream
**CTCs**					Controversial data, scarcity in the bloodstream.Short half-life after blood draw
**EVs**					Need of clinically validated test
**Micro RNAs**					High cost and scarce availability of the test
**DNA methilation**					Less sensitive test
**Histone modification**					Need of further investigation about treatment efficacy
**TP53 (Ab and ctDNA)**					Scarce data from PtR OC
**BRCA (somatic and germinal)**					Reversion mutation
**Immune related biomarkers**					Low specificity, not universally established cut off, scarce data from PtR OC
**Angiogenic markers**					Scarce and controversial data

Abbreviations: Ab Antibodies, BRCA Breast Cancer gene, CTCs Circulating Tumor Cells, ctDNA Circulating Tumor DNA, EVs Extracellular Vesicles, OC Ovarian Cancer, PtR Platinum-Resistant.

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
