# Peer review of "Role of Circulating Biomarkers in Platinum-Resistant Ovarian Cancer"

_ijms, 2021, doi:10.3390/ijms222413650_

Round 1

Reviewer 1 Report

This is an interesting review that is focused on the known biomarkers of ovarian cancer. Treating ovarian cancer patients especially in the cases of progressive resistance to chemotherapy is a really challenging hot topic.  Noninvasive examination, as liquid biomarkers are used for, will be a major component of the emerging field of personalized medicine. Authors collected and systematized a large amount of published data that could be of interest to medical and scientific communities. Text is easy to read and understand.

The manuscript can be accepted for publication when following corrections are made:

  • Chapter 1. Introduction

Are reviewed biomarkers useful for screening all women or only for patients that are known to have ovarian cancer?  Several sentences about this in the introduction part will benefit the manuscript.

  • Tables 1 and 7

These tables should have more visible row separators.

  • Chapter 3.3.2.2 Homologous recombination and BRCA genes

“..such as Poly (ADP-ribose) polymerase inhibitors (PARP is) (predictive role). Indeed, PARPis prevent the mechanism…”

These sentences should be written as “…such as Poly (ADP-ribose) polymerase inhibitor that is termed a PARP inhibitor. Indeed, PARP inhibition prevents the mechanism…”

  • Chapter 4. Conclusions…

Please, change spelling “gyneacology” to “gynecology” as it used overall the manuscript.

Reviewer 2 Report

The author wrote a wonderful review paper focused on circulating biomarkers in platinum resistant ovarian cancer. The authors do a great job introducing and synthesizing the literature for each topic. This systematic review is very well organized and extremely well written. I just have a few minor comments for the authors listed below. 

3.21. Circulating tumor DNA section-

1) The authors mention that there are two modes for ctDNA to be released into circulation: spontaneous lysis and in response to death due to treatment. Later in this section it is cited that increased ctDNA showed resistance to Bevacizumab. Could this be due to when the blood was collected after treatment? The authors did a good job at describing the  the caveats with ctDNA as a biomarker but it would help to emphasize that there could be difficulties in determining if the ctDNA is due to highly proliferating tumors, tumor shedding, ect. vs response to therapies and tumor killing. Also that timing for when the samples are collected could tell very different stories. 

2) Line 197- I think the sentence should read and its metabolism and diffusion. The and is missing.  

3.2.2 Circulating tumor cells

3) It could be assumed that the cells that are released and able to circulate in the tumor are different (more advanced, metastatic in nature, ect.) than many of the tumor cells still in the solid tumors. Could the authors provide some further evidence and description of how the CTCs mirror the heterogeneity of the primary tumor site?

3.3.1.2 DNA methylation

4) Line 332- the citation will need to be corrected.

3.3.1.3 Histone modification

5) Line 421- The s on HDACs should be removed. Line 425- the period after nowadays should be changed to a comma. 

3.3.2.2. HR and BRCA genes

6) Figure 4- The labeling in the figure does not match the figure legend. It states that the tissue samples (somatic mutation) should be 14% and in the  figure it is 20% and similarly for the BRCA in the blood samples. The should be switched. Also, it is misleading that that the figure represents true percentages of the population. Either in the figure title or description it should be stated that these are approximate percentages. Better yet, it could be more reflective of the percentages if the authors move the boxes to include parts of the people images (similarly to how NIH Seer represents percentages).

7) Line 495- The space in PARP is should be removed.

3.5 Immune related biomarkers

8) There  is new exciting genetics data about response to ICIs emerging in the literature from samples analyzed from the NRG-GY003 clinical trial. The data here supports that the fraction of genome altered (FGA) could be a predictor of response while BRCA1/2 mutations, HRD were not (unlike reported in other studies). Genetics also play a role in chemo resistance. Comparisons could also be drawn.  doi: 10.1200/PO.20.00069. 

Conclusions

9) Line 648- "ca ne be obtained" should be changed to "can be obtained"

10) Table formatting should be corrected prior to publication. Expectations are that this would be modified during the final production but I wanted to point it out here.  
